# Synthesis of NiMoO_4_/NiMo@NiS Nanorods for Efficient Hydrogen Evolution Reactions in Electrocatalysts

**DOI:** 10.3390/nano13121871

**Published:** 2023-06-16

**Authors:** Sen Hu, Cuili Xiang, Yongjin Zou, Fen Xu, Lixian Sun

**Affiliations:** 1School of Material Science & Engineering, Guangxi Key Laboratory of Information Materials and Guangxi Collaborative Innovation Center of Structure and Property for New Energy and Materials, Guilin University of Electronic Technology, Guilin 541004, China; guethusen@163.com; 2School of Mechanical & Electrical Engineering, Guilin University of Electronic Technology, Guilin 541004, China

**Keywords:** overpotential, catalysis, electrolyzing water, high-performance materials

## Abstract

As traditional energy structures transition to new sources, hydrogen is receiving significant research attention owing to its potential as a clean energy source. The most significant problem with electrochemical hydrogen evolution is the need for highly efficient catalysts to drive the overpotential required to generate hydrogen gas by electrolyzing water. Experiments have shown that the addition of appropriate materials can reduce the energy required for hydrogen production by electrolysis of water and enable it to play a greater catalytic role in these evolution reactions. Therefore, more complex material compositions are required to obtain these high-performance materials. This study investigates the preparation of hydrogen production catalysts for cathodes. First, rod-like NiMoO_4_/NiMo is grown on NF (Nickel Foam) using a hydrothermal method. This is used as a core framework, and it provides a higher specific surface area and electron transfer channels. Next, spherical NiS is generated on the NF/NiMo_4_/NiMo, thus ultimately achieving efficient electrochemical hydrogen evolution. The NF/NiMo_4_/NiMo@NiS material exhibits a remarkably low overpotential of only 36 mV for the hydrogen evolution reaction (HER) at a current density of 10 mA·cm^−2^ in a potassium hydroxide solution, indicating its potential use in energy-related applications for HER processes.

## 1. Introduction

As the global economy continues to develop, countries are accelerating the research and development of clean energy to address increasingly severe environmental problems [1,2,3,4,5,6]. Hydrogen, as an ideal clean energy source, has received widespread attention. Hydrogen energy produces less pollution compared with traditional fossil fuels; further, hydrogen has more flexible storage and conversion methods and can thus meet the energy consumption requirements for different levels of demand [7,8,9,10]. However, certain technical challenges need to be overcome before hydrogen can be used on an industrial scale for energy generation. Among them, the high energy consumption and cost involved in the hydrogen production process represent significant bottlenecks [11,12,13,14]. To overcome these problems, numerous researchers have begun exploring efficient catalysts for the evolution of hydrogen that are cost-effective for electrical transport. Electrochemical hydrogen evolution is an important method for hydrogen production, and it relies on efficient catalysts to reduce energy consumption and increase the hydrogen production rate [15,16,17,18,19]. Although precious metal catalysts, such as platinum, have high electrochemical activity and stability, their high cost and limited resource reserves limit their widespread application. Therefore, low-cost and efficient nonprecious metal catalysts must be developed [20,21,22].

The excellent overpotential exhibited by transition metal sulfide (TMS) materials has received significant attention [23,24,25,26]. TMS catalysts exhibit good electrochemical catalytic performance and stability [27,28]. The introduction of transition metal elements increases the electrochemically active surface area of the catalyst. Transition metals have unique electronic properties that allow them to form complexes and coordinate with other molecules. This creates more active sites on the catalyst surface for the reactants to interact with, which can lead to a faster and more efficient reaction [29,30]. Moreover, the structure and performance of TMS materials can be optimized by adjusting their preparation methods and conditions, thereby achieving higher catalytic activity and stability [31,32,33]. This provides greater potential for their application in the field of electrochemical hydrogen production. In addition, TMS materials are sustainable and environmentally friendly; thus, they are highly regarded in the field of electrochemical hydrogen production [34]. TMS materials have good prospects for large-scale production owing to their simple preparation process and low cost of raw materials [35,36]. Moreover, their abundant resources, environmental friendliness, and sustainability characteristics meet current requirements for environmental sustainability; thus, they offer significant potential in the field of electrochemical hydrogen production [37]. Franceschini et al. [38]. reported a nanotube catalyst based on nickel-molybdenum sulfide, which has excellent electrochemical hydrogen evolution performance and can achieve efficient hydrogen production at low potentials. Karuppasamy et al. [39] reported that a sulfide catalyst based on molybdenum and cobalt doped, which has high electrochemical hydrogen evolution activity and stability, can achieve efficient hydrogen production under neutral conditions and has a simple preparation method with low cost. These studies indicate that catalysts based on TMS materials have significant potential in the field of electrochemical hydrogen production and also provide novel ideas and methods for future research.

NiS is another material currently being considered for electrochemical hydrogen production [40]. NF was selected as the substrate for the study due to its high porosity and electrical conductivity; however, its bare framework structure may result in an uneven dispersion of active species [41,42]. To enhance rapid electron transport, the incorporation of NiMoO_4_/NiMo onto the NF not only elevates the specific surface area but also preserves the structure.

In this study, a novel electrocatalyst was developed by designing and synthesizing NiMoO_4_/NiMo@NiS-modified NF substrates employing hydrothermal and chemical electroplating methods. Characterization techniques, along with a comprehensive synthesis process, were employed to assess the electrochemical properties and microscopic morphology of the materials both before and after the formation of the composite. A hydrothermal technique was used for the growth of NiMoO_4_ and NiMo layers on the electroplated nanofibers, followed by the deposition of NiS nanoparticles onto the surface. The proposed synthesis method and characterization techniques provide valuable insights into the microscopic morphology and electrochemical properties of the NiMoO_4_/NiMo@NiS-modified NF substrates.

## 2. Materials and Methods

### 2.1. Synthesis of NF/NiMoO_4_/NiMo

The NF surfaces were purified using a sequence of cleaning agents, including a solvent, an acid, a dehydrated alcohol, and distilled water to remove any impurities like oil or oxide layers. The substrates were then dehydrated in a vacuum oven at 60 °C. To synthesize NF/NiMoO_4_/NiMo, nickel nitrate and ammonium molybdate were dissolved in ultrapure water, and the solution was heated in an autoclave at 150 °C for 6 h with the dried NF substrates. After cooling to room temperature, the resulting samples were rinsed with dehydrated alcohol and distilled water multiple times and dried at 60 °C for 12 h, leading to the formation of the NiMoO_4_/NiMo precursor on the NF surfaces.

### 2.2. Synthesis of NF/NiMoO_4_/NiMo@NiS

To prepare the NF/NiMoO_4_/NiMo@NiS sample, the NF was cut into four 1 cm × 2 cm pieces and placed in a clean petri dish for later use. During the electrochemical deposition process, the CHI 760E workstation, comprising a three-electrode system, was utilized. The working, reference, and counter electrodes were composed of NF/NiMoO_4_/NiMo, a saturated silver/silver chloride electrode, and a platinum wire, respectively. The electrolyte solution consisted of approximately 50 mL of 5 mM Ni(NO_3_)_2_·6H_2_O, 5 mM CH_4_N_2_S, and 7.5 mM Na_3_C_6_H_5_O_7_·2H_2_O. To achieve the desired deposition quality, the deposition voltage, time, and stirring speed were optimized to be −0.8 V, 600 s, and 300 rpm, respectively. The method for preparing NF/NiS, chemicals and reagents used in this work, as well as instrument details, can be found in the Supporting Information.

## 3. Results and Discussion

### 3.1. Physical Characterization

The preparation process of the NF/NiMoO4/NiMo@NiS material with rod and spherical structures is depicted in Figure 1. To explore the composition of the material, the prepared sample was analyzed using XRD (Diffraction of X-Rays). In accordance with the results illustrated by Figure 2a,b, the (1 1 1), (2 0 0), and (2 2 0) crystal planes of Ni were observed as high-intensity peaks at 44.4°, 51.8°, and 76.3°, respectively, in the sample. These can be attributed primarily to the NF substrate (PDF#04-0850) [43]. The crystal plane of NiMoO_4_ was identified as (2 4 1) based on the peak observed at 45.74°. (PDF#45-0142) [43,44]. The characteristic peak for the (2 2 0) crystal plane of NiS (PDF#48-1745) appeared at a diffraction angle of 38.29°. In addition, NiMo was found to have diffraction peaks at 65.47° and 78.02° corresponding to the (5 4 0) and (2 5 5) crystal planes [44], owing to the formation of a NiMo phase during the hydrothermal process. Additionally, the formation of NiMo is also beneficial for electron transfer. Our findings show that the composite material made of NF/NiMoO_4_/NiMo@NiS was successfully prepared.

To demonstrate the relevant microstructure, the generated product was characterized using a scanning electron microscope. Figure 3 exhibits the microscopic morphology of all the samples. Figure 3a–c show the process of directly electroplating a layer of NiS on NF. Figure 3a shows the multi-channel pore structure of NF, which is typical of a 3D material and provides a good skeletal structure for the electroplating of NiS and the growth of NiMoO_4_/NiMo. Figure 3b shows the microscopic image of NiS electroplated on foam nickel, and Figure 3c is an enlargement of Figure 3b, which clearly illustrates that the NiS layer on the NF is formed by small NiS spheres stacked together, with a diameter of approximately 20 nm. The dense NiS layer formed on the NF effectively blocks the contact between NF and oxygen in the air, thereby preventing NF from being oxidized. Figure 3d,e display the microscopic images of a layer of NiMoO_4_/NiMo nanoflowers growing on foam nickel using a hydrothermal method. Compared with foam nickel, NF/NiMoO_4_/NiMo significantly increased the specific surface area and the number of active sites on the material; this provides an increased number of active sites for the subsequent growth of NiS. The optimal conditions for the growing NiMoO_4_/NiMo on the surface of NF are a reaction temperature of 150 °C and a time of 6 h [43]. The NiMoO_4_/NiMo structure prepared by this method was stable and uniformly distributed. Figure 3f shows the SEM image of electroplating NiS on NF/NiMoO_4_/NiMo; evidently, some NiS small spheres were homogeneously grown on the surface of NF/NiMoO_4_/NiMo, thus further increasing the contact area between the material and medium, providing more electron transfer channels, and providing good conditions for subsequent electrochemical performance testing. Thus, in conclusion, the NF/NiMoO_4_/NiMo@NiS material was prepared successfully.

TEM was used to further explore the structural properties of the as-prepared NF/NiMoO_4_/NiMo@NiS composite. The TEM image in Figure 4a demonstrated a rod−like structure of the NiMoO_4_/NiMo composite, with a diameter of approximately 100–200 nm, which is consistent with the SEM results. The HRTEM image in Figure 4b revealed lattice stripe spacings of 0.252 nm, corresponding to the (−2 4 1) crystal planes of NiMoO_4_, and 0.214 nm, confirming the presence of (2 2 0) NiS crystallographic planes in the composite. Furthermore, lattice stripe spacings of 0.408 nm and 0.351 nm, matching the (2 5 5) and (5 4 0) crystal planes of NiMo, respectively, were also observed in the same image.

Evidently, the NiMoO_4_/NiMo nanorods were formed of Ni, Mo, and O, and NiS nanoparticles consisting of Ni and S were consistently distributed on the rod−like NiMoO_4_/NiMo materials in Figure 4c, which depicts the EDS elemental mapping of the composites. EDS plots of NF/NiMoO_4_/NiMo and NF@NiS are shown in Appendix A.

To more precisely ascertain the valence states of the elements within the prepared material, these samples underwent XPS analysis to understand their electronic structure. Figure 5a displays the complete spectrum indicating the locations of the C, Ni, Mo, O, and S elements, which corroborates with the EDS testing outcomes. In Figure 5b, the three peaks were positioned at 284.7, 285.7, and 288.9 eV, with the binding energies in accordance with the C-C/C=C, C-O, and O-C=O bonds [43], respectively, which were produced by the growth of NiMoO_4_/NiMo@NiS on NF. The source of these peaks is caused by the addition of disodium citrate dihydrate in electrochemical deposition. The Ni element exhibited six binding energy peaks, as shown in Figure 5c; of these, the energy peaks at 853, 855.8, 873.5, and 877.8 eV belonged to Ni^0^, Ni^2+^, Ni _2p3/2_, Ni _2p1/2_, respectively. The remaining two broad peaks at 861.2 and 882.4 eV were designated as Ni^2+^ satellite peaks [44], respectively. Likewise, the observed results obviously indicate that NiMoO_4_ /NiMo is composed of different valence states, including Ni^2+^, Mo^6+^, Ni^0^ and Mo^0^ [43,44,45,46], and these groups are tightly linked to create the alloys NiMoO_4_ and NiMo. The Mo peak values in the Mo 3d spectrum were found at 232.2 and 235.3 eV. In addition, corresponding low−valence Mo^0^ and Mo^4+^ were generated within the sample, with respective binding energies of 237.4 eV and 226.8 eV. The binding energy at 237.4 eV increased, possibly because the electrons lost energy during the synthesis process, thus resulting in the kinetic energy of the electrons detected by the instrument being less than the intrinsic energy; consequently, this led to an increase in the binding energy. The binding energies of the oxygen elements were analyzed, with O 1 (530 eV), O 2 (530.9 eV), and O 3 (532.2 eV). The O^2−^ forming oxide with metal elements was affiliated with O 1. The O 2 component can be attributed to the defect sites, and the O 3 component can be summarized as the physi- and chemisorbed water [47,48] component to the physical and chemical adsorption of water. Ni−S bonds are evident because the spectra of S _2p_ reveal peaks at 162.6 and 163.75 eV that can be ascribed to S _2p3/2_ and S _2p1/2_ [49], respectively. Due to the oxidized valence state of the NiS surface, the peaks at 168.5 and 169.5 eV can also be attributed to S−O bonds. The XPS analysis results indicate that the NiMoO_4_/NiMo catalysts activate the O−H bonds in the reactant molecules through the strong coupling between Ni and Mo species, whereas the existence of S and O on the surface of the catalysts enhances the durability and catalytic action by facilitating the formation of active sites and keeping the catalysts from losing their effectiveness. The additive effects of Ni, Mo, S, and O in the prepared catalysts play an important part in the catalytic reaction, which is essential for the design and optimization of catalysts in various chemical reactions. These findings can contribute to the evolution of high−performance catalysts for sustainable chemical processes and energy conversion applications. Further studies are required to elucidate the detailed mechanism of the catalytic reaction and optimize the catalyst design for enhanced performance and stability.

### 3.2. Electrochemical Characterization

The electrocatalytic performance of NF, NF@NiS, NF/NiMoO_4_/NiMo, and NF/NiMoO_4_/NiMo@NiS samples was evaluated for the hydrogen evolution reaction (HER). LSV (Linear Sweep Voltammetry) curves obtained from evaluating these samples in a 1.0 M KOH solution are displayed in Figure 6a. Evidently, the HER activity of all the samples was surpassed by the NF/NiMoO_4_/NiMo@NiS samples. The application of a current density of 10 mA·cm^−2^ necessitated an overpotential of 36 mV; it was second only to Pt/C (24 mV) in terms of the magnitude of the overpotential required. The properties of other materials are shown in Appendix A. Meanwhile, NF, NF/NiMoO4/NiMo, and NF@NiS required overpotentials of 287, 322, and 120 mV, respectively, to achieve the same current densities, and the required overpotentials for each sample were compared. These findings suggest that NF/NiMoO_4_/NiMo@NiS exhibited superior activity compared to all other samples.

The Tafel slopes for each sample were calculated from their respective LSV curves, as demonstrated in Figure 6b, following analysis. The Tafel slope of NF/NiMoO_4_/NiMo@NiS, which was 40.16 mV·dec^−1^, was notably flatter than that of the other samples, including NF, NF/NiMoO_4_/NiMo, NF@NiS, and NF/NiMoO_4_/NiMo@NiS, which displayed Tafel slopes of 117.43, 152.30, and 81.79 mV·dec^−1^, respectively. These findings suggest that NF/NiMoO_4_/NiMo@NiS exhibited faster reaction kinetics during HER compared to the other samples. We think that the Volmer-Heyrovsky reaction may be occurring here.

In order to gain a deeper understanding of the kinetic mechanisms occurring at the electrode surfaces, the Nyquist plots for NF, NF/NiMoO_4_/NiMo, and NF@NiS were obtained using EIS with an open-circuit voltage. Figure 6c shows that NF/NiMoO_4_/NiMo@NiS had a quicker rate of interfacial charge transfer compared to the other samples, as indicated by the Nyquist plots. Through impedance fitting, we obtained the electrical resistance value of NF is 4.56 Ω, and the electrical resistance value of NF/NiMoO_4_/NiMo@NiS is 0.1042 Ω.

The catalytic activity of a material is a crucial factor that determines its effectiveness in various applications. The electrochemically active surface area (ECSA) of a material is a key parameter that affects its catalytic activity, as it directly correlates with the number of available active sites for the catalytic reaction. The ECSA is closely related with respect to the electrochemical of material double-layer capacitance (C_dl_), which is a measure of its ability to store electrical charge at the interface between the solid and liquid phases. The higher the C_dl_ value, the greater the available surface area for catalytic reactions. Therefore, the ECSA and C_dl_ values must be determined to understand and optimize the catalytic performance of materials in several electrochemical applications, for instance, storage and conversion, electrochemical sensing, and wastewater treatment. Further, the reasons behind the improved electrocatalytic activity of the proposed material must be investigated. To determine the value of C_dl_, CV scans were performed within the nonredox potential range. The current density at the CV scan interval’s midpoint was determined from the positive and negative scans and utilized to estimate Δj. The slope of the plot of Δj against the scan rate was found to be a factor of two of C_dl_. Thus, a higher C_dl_ value indicates a greater ECSA and enhanced catalytic activity. Hence, the proposed material’s higher Cdl value explains its improved electrocatalytic activity. This characteristic makes it a potential candidate for diverse electrochemical applications. As demonstrated in Figure 7a, The C_dl_ value of NF/NiMoO_4_/NiMo@NiS was determined to be 6.52 mF·cm^−2^. The CV plots for the other pairs of samples are shown in Appendix A. Thus, the proposed rod-shaped morphology of NF/NiMoO_4_/NiMo@NiS exhibited an exceptionally high ECSA value, suggesting that it could offer a significant number of HER catalytic active sites for the electrocatalytic process.

Stability is a crucial factor in assessing electrocatalysts with high efficiency as it determines performance longevity. To provide a more comprehensive understanding of the stability of NF/NiMoO_4_/NiMo@NiS as an electrocatalyst for HER, several tests were conducted. The current-time plots recorded at a bias voltage of η = 36 mV revealed that the current density of NF/NiMoO_4_/NiMo@NiS exhibited a slight decay after a durability test of up to 10 h. This observation indicates that the proposed material is highly stable even after prolonged exposure to harsh electrochemical conditions. Moreover, to further evaluate the stability of NF/NiMoO_4_/NiMo@NiS, 1000 cycles of CV testing were performed, and at 10 mA cm^−2^, the overpotential barely rose by 9 mV. The stabilized form is shown in Appendix A. This result indicates that the proposed material exhibits remarkable stability under cycling conditions, even at high current densities. Thus, the combined results of both the durability and CV tests clearly demonstrate the superior stability and durability of NF/NiMoO_4_/NiMo@NiS, which is a highly desirable property for electrocatalysts used in HER. These findings suggest that NF/NiMoO_4_/NiMo@NiS is a promising candidate for the practical utilization of various renewable devices for converting and storing energy.

## 4. Conclusions

This study introduced a simple hydrothermal and electrochemical deposition method to grow rod-like NiMoO_4_/NiMo on NF substrate, followed by electroplating NiS nanospheres to prepare NF/NiMoO_4_/NiMo@NiS. The presence of numerous pore structures in the NF significantly enhances the reaction area, thereby providing favorable conditions for the subsequent growth of NiMoO_4_/NiMo, whereas the modification of rod-like NiMoO_4_/NiMo not only improved the electrical conductivity of NF but also provided a larger surface area and excellent surface functional properties. Finally, the deposition of a layer of NiS particles on top of the precursor further enhances the material’s electrical conductivity and exposes more active sites, leading to a decrease in the overpotential required for the hydrogen evolution reaction. The NF/NiMoO_4_/NiMo@NiS catalyst provides a certain amount of defects, which allows H^+^ and e^−^ to easily aggregate on the catalyst, promoting the reduction of H^+^ ions, thus reducing the energy required for the HER reaction. This results in excellent electrocatalytic performance, with an overpotential of only 36 mV at 10 mA·cm^−2^, a low Tafel slope of 40.16 mV·dec^−1^, and excellent cycling stability in a 1.0 M potassium hydroxide solution.

## Figures and Tables

**Figure 1 nanomaterials-13-01871-f001:**
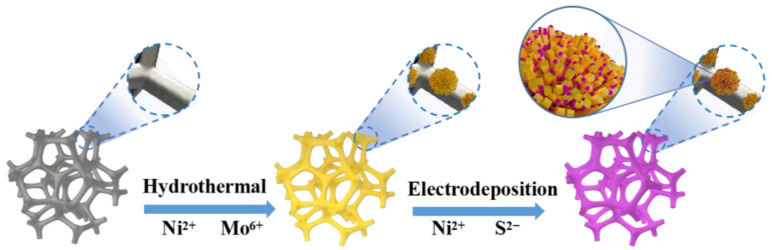
Schematic depicting the synthetic process of the NF/NiMoO_4_/NiMo@NiS spherical−coated rod structure material.

**Figure 2 nanomaterials-13-01871-f002:**
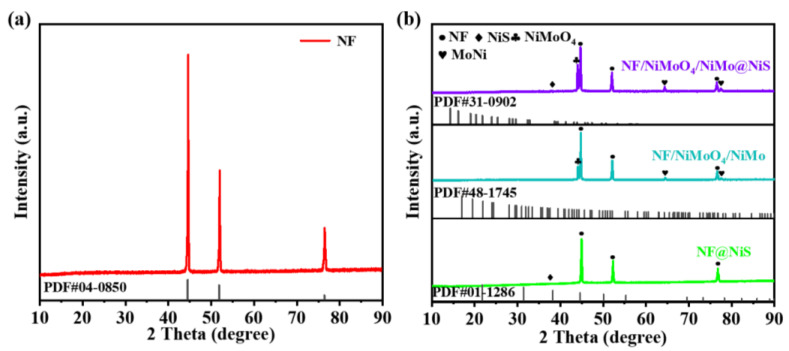
XRD patterns for (**a**) NF and (**b**) NF@NiS, NF/NiMoO_4_/NiMo, and NF/NiMoO_4_/NiMo@NiS samples.

**Figure 3 nanomaterials-13-01871-f003:**
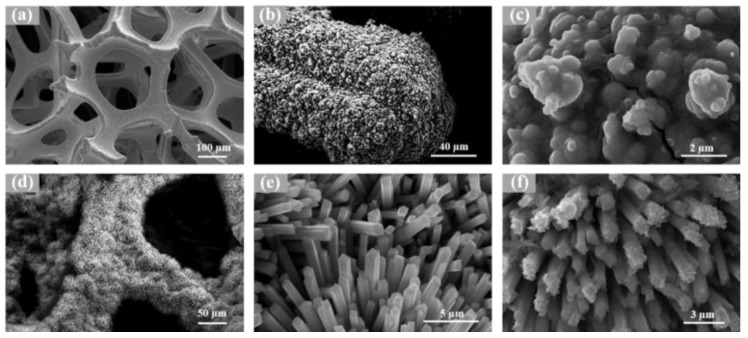
SEM images of (**a**) NF; (**b**,**c**) NF@NiS; (**d**,**e**) NF/NiMoO_4_/NiMo; and (**f**) NF/NiMoO_4_/NiMo@NiS.

**Figure 4 nanomaterials-13-01871-f004:**
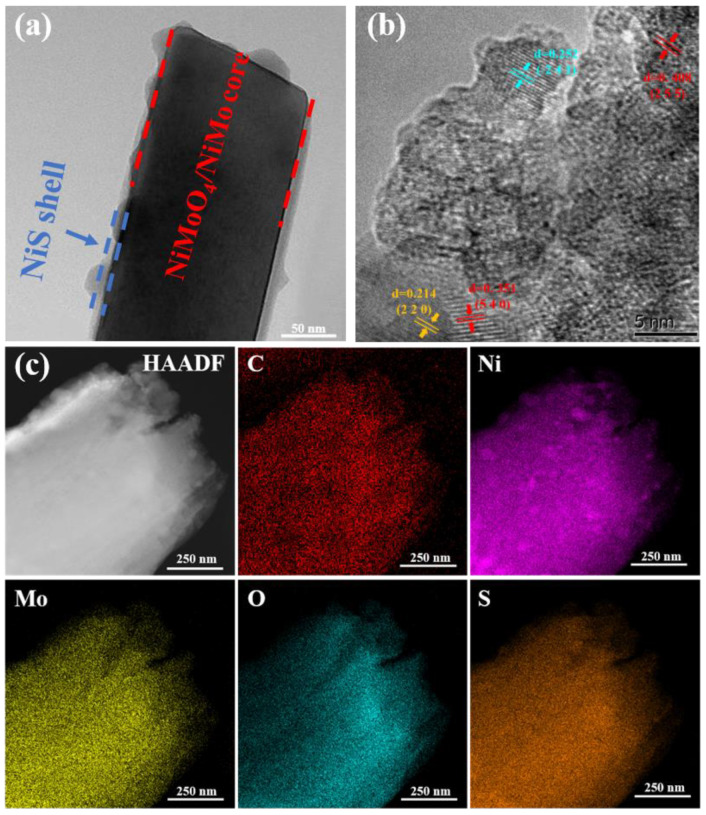
(**a**) TEM images of NF/NiMoO_4_/NiMo@NiS. (**b**) HRTEM pattern and (**c**) EDS spectrum of NF/NiMoO_4_/NiMo@NiS.

**Figure 5 nanomaterials-13-01871-f005:**
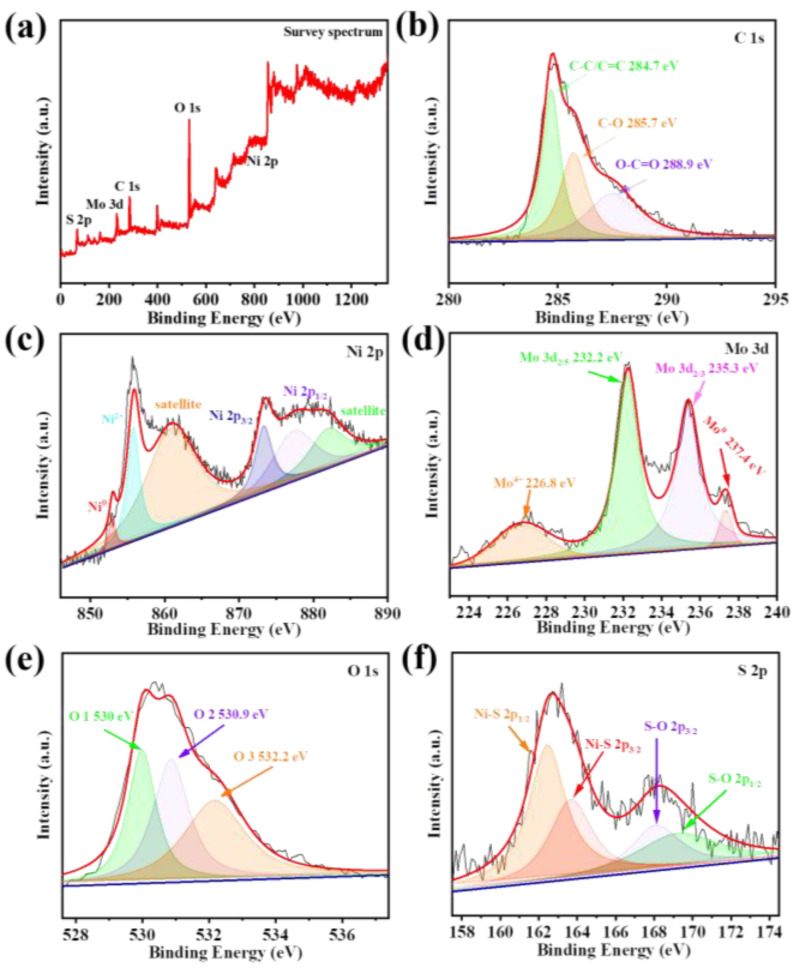
(**a**) For the NF/NiMoO_4_/NiMo@NiS material, the XPS survey spectrum. Spectra of: (**b**) C 1s, (**c**) Ni 2p, (**d**) Mo 3d, (**e**) O 1s, and (**f**) S 2p, in the NF/NiMoO_4_/NiMo@NiS material.

**Figure 6 nanomaterials-13-01871-f006:**
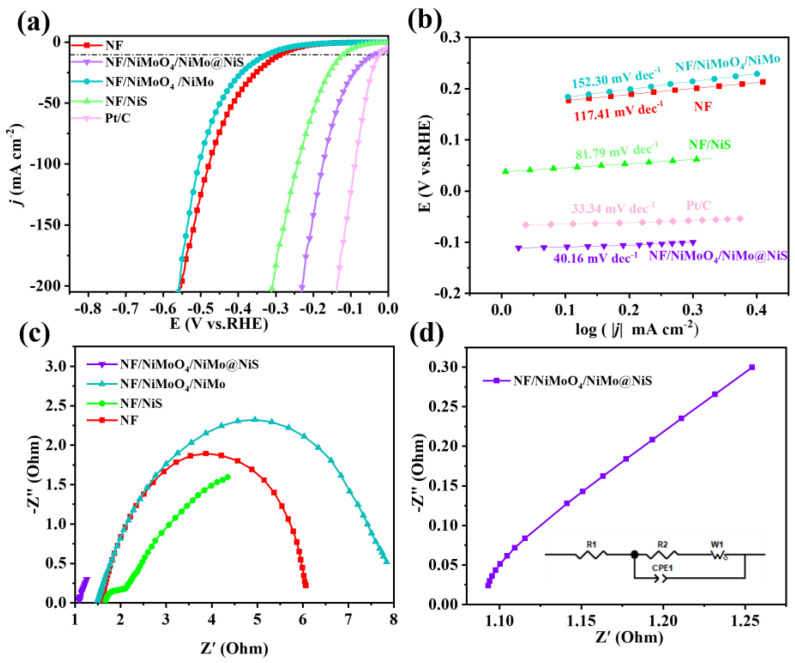
(**a**) Polarization LSV curves for the HER at a scan rate of 5 mV·s^−1^. (**b**) Tafel plots of NF, NF@NiS, NF/NiMoO_4_/NiMo, and NF/NiMoO_4_/NiMo@NiS. (**c**) EIS of NF, NF@NiS, NF/NiMoO_4_/NiMo, and NF/NiMoO_4_/NiMo@NiS in the frequency range between 100 kHz and 10 MHz. (**d**) NF/NiMoO_4_/NiMo@NiS in the frequency range between 100 kHz and 10 MHz. Be aware that 1.0 M KOH solution is used for all electrochemical measurements of the HER performance for the catalysts.

**Figure 7 nanomaterials-13-01871-f007:**
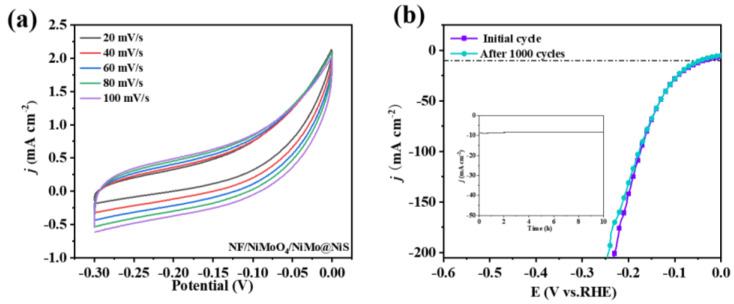
CVs of (**a**) NF/NiMoO_4_/NiMo@NiS measured at different scan rates. (**b**) Initial overpotential and overpotential after 1000 cycles and the 10 h chronopotentiometric curve obtained for NF/NiMoO_4_/NiMo@NiS at a constant current density of 10 mA·cm^−2^.

## Data Availability

Not applicable.

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
