# Peer review of "Synthesis of NiMoO4/NiMo@NiS Nanorods for Efficient Hydrogen Evolution Reactions in Electrocatalysts"

_nanomaterials, 2023, doi:10.3390/nano13121871_

Round 1

Reviewer 1 Report

Please find comments in the attached document.

Quality of English is good, except in some paragraphs mentioned in the attached document.

Author Response

Thank you for your valuable comments. The relevant issues have been revised in the Manuscript and supporting Information.

Reviewer 2 Report

Authors reported the Synthesis of NiMoO4/NiMo@NiS Nanorods for Efficient Hydrogen Evolution Reactions in Electrocatalysts. The NF/NiMo4/NiMo@NiS material exhibits a remarkably low overpotential of only 36 mV for  the hydrogen evolution reaction (HER) at a current density of 10 mA·cm-2. The NF/NiMo4/NiMo@NiS demonstrates HER activity of NF/NiMo4/NiMo@NiS very close to commercial Pt-C. following revisions should be made before publication.

1.      Experiments have shown that the addition of appropriate materials can enhance the physicochemical properties of the material. What is the meaning of enhancement of the physicochemical properties? Abstract should be rearranged.

2.      In NF/NiMo4/NiMo@NiS, does NiMo phase consists metal alloy phase? If so, how can you prove it. The metal peaks present may be due to the Ni foam. Compare it with the XRD of Ni foam alone and NiMo alone.

3.      In introduction section, the rational use of metal sulfides/oxide electrodes can be explained in detail with the reference of following articles:

doi.org/10.1016/j.jallcom.2023.170678, doi.org/10.1016/j.est.2023.106713, doi.org/10.1016/j.compositesb.2022.110339

4.      NF/NiMo4/NiMo@NiS, which component has major role for the excellent HER activity.

5.      How about the morphology after stability. Also, compare the tafel plots for Pt-C.

6.      The mechanism of HER process should be explained in detail.

7.      The author's explanation of the reason for the performance is very vague and insufficient. The author should analyze the real reason for improving the catalytic performance more deeply and adequately.

Some English correction is necessary. Some sentences are arranged improperly.

Author Response

(The authors gave the same response as above.)

Round 2

Author Response

Thank you very much for your comments, we have made the requested changes to the supporting Information.

Reviewer 2 Report

Accept

Author Response

Thanks to the reviewers and editors for their help in revising and publishing the article.